# Next-Generation Human Liver Models for Antimalarial Drug Assays

**DOI:** 10.3390/antibiotics10060642

**Published:** 2021-05-27

**Authors:** Kasem Kulkeaw

**Affiliations:** Department of Parasitology, Faculty of Medicine Siriraj Hospital, Mahidol University, Bangkok 10700, Thailand; kasem.kuk@mahidol.edu; Tel.: +662-419-6468 (ext. 96484)

**Keywords:** malaria, plasmodium, antimalarial drug, hepatocyte, liver/hepatic organoid, pluripotent stem cell

## Abstract

Advances in malaria prevention and treatment have significantly reduced the related morbidity and mortality worldwide, however, malaria continues to be a major threat to global public health. Because *Plasmodium* parasites reside in the liver prior to the appearance of clinical manifestations caused by intraerythrocytic development, the *Plasmodium* liver stage represents a vulnerable therapeutic target to prevent progression. Currently, a small number of drugs targeting liver-stage parasites are available, but all cause lethal side effects in glucose-6-phosphate dehydrogenase-deficient individuals, emphasizing the necessity for new drug development. Nevertheless, a longstanding hurdle to developing new drugs is the availability of appropriate in vitro cultures, the crucial conventional platform for evaluating the efficacy and toxicity of drugs in the preclinical phase. Most current cell culture systems rely primarily on growing immortalized or cancerous cells in the form of a two-dimensional monolayer, which is not very physiologically relevant to the complex cellular architecture of the human body. Although primary human cells are more relevant to human physiology, they are mainly hindered by batch-to-batch variation, limited supplies, and ethical issues. Advances in stem cell technologies and multidimensional culture have allowed the modelling of human infectious diseases. Here, current in vitro hepatic models and toolboxes for assaying the antimalarial drug activity are summarized. Given the physiological potential of pluripotent and adult stem cells to model liver-stage malaria, the opportunities and challenges in drug development against liver-stage malaria is highlighted, paving the way to assess the efficacy of hepatic plasmodicidal activity.

## 1. Global Impact of Malaria and Urgent Needs

Advances in malaria prevention and treatment have significantly reduced its worldwide morbidity and mortality, however, malaria continues to be a major threat to global public health as the estimated numbers of malaria cases and deaths have not changed substantially since 2010. From 2010–2019, more than 200 million cases of malaria and more than 400,000 deaths were reported worldwide [1]. The causes of malaria are protozoan parasites, including *Plasmodium falciparum*, *P. vivax*, *P. ovale*, *P. malariae* and *P. knowlesi,* among which *P. falciparum* is the most virulent and dominant species, and *P. vivax* has the widest geographical distribution and major economic impact. According to the World Malaria Report, the majority of malaria cases in 2018 occurred in Africa (93%), followed by the South-East Asia (3.4%) and the Eastern Mediterranean (2.1%). Children less than 5 years of age have the highest morbidity and death risks due to malaria [2]. Therefore, global eradication of malaria is warranted and a campaign to achieve this has been launched.

Given that *Plasmodium* parasites reside in the liver prior to intraerythrocytic development, the *Plasmodium* liver stage represents a vulnerable target for therapeutic interventions to block progression to clinical malaria and to prevent relapse in subjects with vivax and ovale malaria. Currently, atovaquone and proguanil are prophylactic regimens that specifically target liver-stage schizonts, while primaquine and tafenoquine, a group of 8-aminoquinolines, are capable of killing hypnozoites [3]. To prevent relapses of vivax malaria, a full 14-day course of oral primaquine administered daily is required. Although two recent reports showed the effectiveness of a single dose of tafenoquine as a radical cure in a phase-3 clinical trial [4,5], primaquine and tafenoquine could cause hemolysis in G6PD-deficient patients. Thus, its cytotoxic effect limits mass administration in at-risk patients. Collectively, these unsolved, long-known situations lead to the need for safer therapeutics that effectively hinder the development of *Plasmodium* parasites in the liver.

Along the pipeline to find drugs that act as radical cures, in vitro cell culture is an essential preclinical tool for testing the efficacies and toxicities of potential drugs prior to clinical phases. A long-standing hurdle in the preclinical phase is the lack of an in vitro model capable of recapitulating the complex biological structure and functions observed in humans. Thus, two-dimensional (2D) cultured cells represent the leading cause of failure in clinical studies in humans [6]. In this review, advances in physiologically relevant models of malaria as well as the potential of human-induced pluripotent stem cells (iPSCs) are highlighted.

## 2. Intrahepatic Development of Plasmodium Parasites: An Unfilled Gap in Malaria Biology

At present, five *Plasmodium* species, *P. falciparum*, *P. *vivax**, *P.* ovale, *P. *malariae** and zoonotic *P. knowlesi*, cause human malaria. All *Plasmodium* species develop in different biological niches. In *Anopheles* mosquitoes, they undergo sexual development, but they change to asexual development in humans. Intrahepatic development of *Plasmodium* parasites causes no illness, whereas intraerythrocytic development leads to severe clinical symptoms: cerebral malaria, multiorgan complications and anemia. The mechanism underlying *Plasmodium* growth and development in erythrocytes is well known, owing to advances in their laboratory culture. In contrast, the intrahepatic development of malaria parasites remains poorly understood. As shown in Figure 1 [7], *Plasmodium* sporozoites are inoculated with a female mosquito, and parasites reach the liver via circulation. In the liver, sporozoites translocate across liver sinusoid endothelial cells prior to invading hepatocytes in the liver parenchyma. Some studies have suggested that sporozoites must first invade resident macrophage Kupffer cells before entering hepatocytes. However, others have demonstrated the direct invasion of hepatocytes by sporozoites. Thus, the mechanism underlying hepatocyte invasion remains debatable (see extensive reviews by Ejigiri and Sinnis [8] and Deslyper et al. [9]). In the liver parenchyma, hepatocyte-residing sporozoites subsequently invade adjacent hepatocytes. Upon arriving at a given hepatocyte, sporozoites start to develop into merozoites, which then rapidly multiply several thousand times to form schizonts, a process termed schizogony. Following the rupture of schizonts, tens of thousands of merozoites egress from the infected hepatocytes, enter blood circulation, and infect erythrocytes. In infected erythrocytes, merozoites undergo intraerythrocytic schizogony: they develop into ring-formed trophozoites, trophozoites and schizonts. Schizonts repeatedly rupture and release merozoites to start another round of intraerythrocytic progression (Figure 1). Many questions remain unanswered, such as why sporozoites need to invade adjacent hepatocytes after initial invasion, which factor regulates the transition of sporozoites to schizonts and how thousands of merozoites break the host cell membrane to enter the blood circulation. Notably, *P. vivax* and *P. ovale* may remain dormant in a metabolically inactive stage called hypnozoites. The mechanisms underlying the formation of hypnozoites remain unknown. The hypnozoites can be reactivated, resulting in clinical relapse months or years later. A study in an endemic area of Africa showed that *P. vivax* malaria was likely caused by hypnozoites, suggesting the potential of transmission reservoirs. Thus, relapsing *P. vivax* malaria also causes significant clinical and financial burdens [10] and remains challenging for global malaria eradication programs. Collectively, understanding parasite development within liver niches is crucial for the development of methods to prevent the clinical development of malaria and relapse.

## 3. Limitations in the Treatment of Liver-Stage Malaria

Antimalarial drugs target intraerythrocytic and intrahepatic stages. Artemisinins are the only effective front-line drug to treat intraerythrocytic falciparum malaria and other human malarias. Artemisinin-based combination therapy is recommended for the treatment of *P. falciparum* malaria, whereas chloroquine is recommended for the treatment of vivax malaria. Artemisinin-based compounds are combined with a drug from a different class. Artemisinin derivatives include dihydroartemisinin, artesunate and artemether, and the companion drugs include lumefantrine, mefloquine, amodiaquine, sulfadoxine/pyrimethamine, piperaquine and chlorproguanil/dapsone [7].

Currently, numerous treatments targeting the liver stage of the parasite life cycle are inadequate. Prophylactic regimens of atovaquone and proguanil target only liver-stage schizonts, while the only interventions currently capable of targeting hypnozoites are 8-aminoquinolines, such as primaquine and tafenoquine [4,5]. To prevent relapse of vivax malaria caused by hypnozoites in the liver, a full 14-day course of primaquine is effective [3]. However, the daily administration of primaquine can cause dangerous hemolysis in G6PD-deficient patients. Another major problem in malaria treatment is an increase in drug resistance. Cases of *P. falciparum* resistance to artemisinins have been detected in five countries in the Greater Mekong subregion: Cambodia, Lao People’s Democratic Republic, Myanmar, Thailand and Vietnam. Moreover, cases of chloroquine-resistant *P. vivax* malaria have been confirmed in 10 countries: Bolivia, Brazil, Ethiopia, Indonesia, Malaysia, Myanmar, Papua New Guinea, Peru, the Solomon Islands and Thailand. Resistance to both chloroquine and sulfadoxine–pyrimethamine, formerly effective antimalarial drugs, emerged in Western Cambodia and at the Thailand–Myanmar border [7]. In both cases, the resistance genes spread to Africa and caused millions of deaths. Thus, once artemisinin-resistant *P. falciparum* spreads to other parts of the world, new effective drugs are urgently needed.

## 4. Current In Vitro Hepatic Models for Human Malaria and Unaddressed Issues

Various forms of conventional 2D culture have been used to model the intrahepatic development of *Plasmodium* parasites, allowing the assessment of antimalarial drugs and new chemicals (Table 1). At present, most 2D culture systems involve the maintenance of hepatoma-derived cells, primary human hepatocytes (PHHs) or genetically immortalized hepatocytes. There are two lines of hepatoma-derived cells, HepG2-A16 and HHS-102, that have been evaluated as models for liver-stage malaria. Despite the lack of CD81 on the surface of HepG2 cells [11], sporozoites of *P.*
*vivax*, but not *P. falciparum*, were able to invade HepG2-A16 cells [12] and to develop into schizonts capable of releasing merozoites [13]. In contrast, the HHS-102 hepatoma cell line was susceptible to a Thai isolate of *P.*
*falciparum* sporozoites. By day 12 after sporozoite inoculation, erythrocyte-infected merozoites were microscopically detected [14]. Given that they are a general feature of cancer, hepatoma-derived cells exhibit high proliferative activity, a characteristic that differs from that of dormant hepatocytes in the human liver under homeostasis. Therefore, overgrowth of hepatoma cells eventually leads to cell detachment from a plastic plate, restricting the use of long-term culture to study hypnozoites of *P. vivax*. Although cell proliferation can be controlled using chemicals or radiation [13], this process is complicated and may affect the intrahepatic development of *Plasmodium* spp.

Compared to hepatic carcinoma cells, PHHs cultured in 2D are a more physiologically relevant model, and they are commercially available. In a pioneering study on the use of PHHs, *P. falciparum* sporozoites were shown to develop from schizonts into erythrocyte-infective merozoites in 2D PHH cultures [16]. By using PHH-based culture, surface receptors (CD81 [23] and SR-BI [24]) and the molecular mechanism underlying sporozoite invasion into hepatocytes have been revealed [12,25,26]. However, the sporozoite invasion rate in PHHs varies depending on the PHH origin [17,18] and on the *Plasmodium* parasite strain [27]. The 2D platform was further advanced by the microscale coculture of PHHs together with murine embryonic fibroblasts. Both *P. falciparum* and *P. vivax* sporozoites were shown to complete intrahepatic development, producing infective merozoites [17]. Owing to the ability of long-term culture, presumptive hypnozoites of *P. vivax* formed in this PHH fibroblast culture. To develop a successful platform for high-throughput screening, a recent study aimed to establish a PHH culture on 384-well collagen-coated plates. Extensive screening of PHH sources revealed that this platform allows for complete *Plasmodium* development, hypnozoite formation, antibody-based inhibition of sporozoite invasion and evaluation of prophylactic and radical cure drugs [18]. Although 2D-based PHH cultures have been used to study liver-stage malaria, variations in the *Plasmodium* susceptibility of PHH lots [19], the limited number of donors and complicated culture steps are major drawbacks of the PHH platform. Thus, the terminally differentiated PHHs and mesenchymal cells were genetically transformed into immortalized cell lines, thereby providing less quality variation, a constant supply and scalability. HC-04 cells, for instance, could support the complete development of *P. falciparum and P. vivax*, as they were capable of releasing hepatic merozoites capable of infecting human erythrocytes [19]. Thus, a tool to investigate the surface receptors of *P. falciparum* sporozoites was deployed [20]. March et al. reported similar *P. falciparum* infection rates between fibroblast-cultured PHHs and HC-04 cells [17]. However, a comparison of cryopreserved PHHs and HC-04 cells showed the former was superior to HC-04 cells regarding the intracellular *P. falciparum* yield [28]. In addition, immortalized hepatic cells (imHCs), genetically transformed mesenchymal cells (CD90+ CD105+ CD34- CD45-) [29], possess a *Plasmodium* infectivity ability similar to that of HC-04 cells. By contrast, the imHCs were not overgrown upon reaching confluence, allowing maintenance in culture for up to 6 weeks without multilayer formation and cell detachment [21]. However, whether *P. falciparum*-infected imHCs can produce merozoites capable of infecting erythrocytes, an indicator of complete intrahepatic development, needs to be determined.

To overcome the donor shortage and batch-to-batch variation of PHHs, an expandable, less variable hepatocyte source is emerging as an alternative. Human iPSCs are expandable and capable of differentiating into hepatocytes and can be generated from individual donors who have different susceptibilities to *Plasmodium* infection; thus, screening donors with high permissibility to infection will overcome batch-to-batch variation. To date, a protocol for the generation of hepatocytes from human iPSCs is well established [30,31]. Human iPSC-derived hepatocytes are susceptible to hepatitis C [32] and hepatitis B virus [33], major hepatotropic pathogens. A recent study reported the development of a 2D coculture system that consisted of human iPSC-derived hepatocytes and stromal cells. Ng et al. clearly showed that human iPSC-derived hepatocytes express CD81 and SR-BI, cell receptors of *Plasmodium* sporozoites. Although *P. falciparum* sporozoites were shown to fully develop into merozoites in this system [22], the invasion rate of sporozoites remains unexamined. For a more complex model relevant to the human liver, mice engrafted with PHHs and humanized mice were deployed. Although malaria parasites could develop in this human-mouse chimeric model, its high cost and low throughput hinder its use for drug discovery [34,35,36]. Altogether, the availability, scalability and maturity of hepatocytes are key constraints in modeling liver-stage malaria.

## 5. Current Toolbox for Assaying the Plasmodicidal Activity of Liver-Stage Malaria

Hepatocytes in the human liver contain several drug metabolism enzymes, among which CYP2D6, CYP3A4, CYP2C19, and MAO-A reportedly metabolize primaquine [37], resulting in a bioactive primaquine metabolite [38,39,40]. Thus, prior to testing a drug or small molecule against liver-stage malaria, cytochrome P450 activity is assessed. March et al. developed a medium-throughput system based on the microscale coculture of PHHs with murine fibroblasts in 96-well plates. The micropatterned PHHs expressing mRNAs encoding primaquine-metabolizing enzymes (*MAO-A*, *CYP3A4* and *CYP2C19*) were more capable of metabolizing primaquine than the HC-04 cells and monocultured PHHs. Notably, the micropatterned coculture was maintained for more than 3 weeks without the loss of hepatocyte functions (albumin and urea production and CYP450 activity) [17,41], allowing an antihypnozoitidal assay. Although small *P. vivax* forms detected in the micropatterned coculture were presumably hypnozoites, their reactivation needs to be further examined (Table 2).

To develop a high-throughput screening platform, Roth et al. cultured PHHs in a 384-well plate. Together with high-content imaging, this platform allows for the assessment of schizonticidal and hypnozoticidal activity using a relatively low number of hepatocytes and sporozoites [17]. As a demonstration of small-molecule screening, 36 compounds from the Medicine for Malaria Venture (MMV) database and 913 repurposed compounds from the Calibr Bioactive Library were screened. Because these compounds do not require metabolism in hepatocytes, the screening results were devoid of false negatives. By contrast, primaquine must be metabolized by CYP2D6 to produce an active primaquine metabolite. Thus, prescreening of PHH lots for CYP2D6 activity is necessary. Given that the byproduct of primaquine metabolism is unquantifiable, batch-to-batch PHH variability remains a major obstacle, emphasizing the need for the phenotypic screening of cryopreserved hepatocytes (Table 2).

As another hepatocyte source, imHCs were shown to express *CYP3A4* and *CYP2D6* mRNA at higher levels than HC-04 cells. Thus, liver schizonts and small forms of *P. vivax* were reasonably sensitive to primaquine. Regarding iPSC-derived hepatocytes, given the low level of drug metabolism enzymes relative to PHHs [37], *Plasmodium* parasites in iPSC-derived hepatocytes respond only to atovaquone, which does not require hepatic activation [42]. By contrast, since primaquine activation requires hepatic enzymes [43], it failed to inhibit the growth of *Plasmodium* parasites in iPSC-derived hepatocytes. Therefore, the maturity of hepatocytes derived from iPSCs is critical for the assessment of drug efficacy.

In addition to human malaria, attempts to identify plasmodicidal drugs also relied on rodent malaria species, *P. yoelli* or *P. berghei*. Rodent malaria models were used to screen a large-size compound library (>500,000) using a 2D culture of human hepatoma cells with luciferase-expressing *P. berhhei* [44]. Highly sensitive imaging of bioluminescence allows the detection of developing *P. berghei* in mouse liver. In combination with the 2D culture of murine hepatocytes, screening of more than 4000 blood stage-inhibiting compounds led to the discovery of a main scaffold of compounds active against both liver- and blood-stage malaria in rodents [45]. Given the lack of hypnozoites, rodent malaria does not allow antirelapse drug testing. However, humanized mice (NOD-*scid IL2R**γ^null^*) were capable of modeling liver-stage falciparum malaria, leading to the discovery of a new drug inhibiting phosphatidylinositol 4-kinase with the potential for both radical cure and prophylaxis [46]. Notably, the metabolism of the human and mouse liver differ. Thus, any antiplasmodial drug against rodent malaria still needs to be examined in a human context.

## 6. Three-Dimensional Cell Culture Is Emerging as a Next-Generation Malaria Model

Conventionally, cells are cultured as a monolayer on plastic polystyrene or glass surfaces with/without a protein coating. Thus, each cell is able to interact with other nearby cells via lateral cell-cell junctions and the basal extracellular matrix (ECM), which are the two dimensions (2D) of cell culture. Because all cells are equally exposed to nutrients and gas in 2D culture medium, their microenvironments are distinct from those of cells in tissues, where each cell type interacts with the same or different types in all directions (referred to as 3D) and is exposed to nutrients and gas in a gradient manner [47]. In a 3D culture, cell behaviors regarding exogenous stimulus responses, cell signaling, the gene expression profile, and metabolism are reportedly different from those in 2D culture but relatively similar to those in vivo [47]. Therefore, many attempts to recapitulate complex microenvironments in human tissues have been made based on 3D models, including spheroids, organoids and cell/tissue-on-chip systems. In this section, the applications of hepatic spheroids for malaria studies and the potential of hepatic organoids are discussed.

### 6.1. Spheroids

Spheroids are 3D cell clusters generated by the forced aggregation of cells by means of stirring, sedimentation or gravity. Typically, spheroids are derived from cell lines and thus contain only a single cell lineage. Two reports were recently published on the use of human spheroids for testing antimalarial drugs.

#### 6.1.1. Human Hepatoma and Immortalized Hepatic Cell-Derived Spheroids

Hepatoma HepG2 cells and immortalized HC-04 cells were cultured separately in a stirring tank to form spheroids. Then, the human hepatoma spheroids were subjected to sporozoite invasion using a static or dynamic (stirring) system. Both HepG2- and HC-04-derived spheroids could be infected by *P. berghei*. However, the *P. berghei* infection rate in the 2D static format did not differ from that in the 3D dynamic format based on detection of the green florescent protein, a surrogate marker of intracellular parasites. In human hepatoma spheroids, the rodent malaria parasite *P. berghei* could fully develop into liver schizonts in the 2D static and 3D stirring models. However, only the HC-04 -based spheroids could release merozoites capable of infecting mouse erythrocytes. Given the luciferase strain of *P. berghei*, HepG2-based spheroids could be used for drug testing [48].

#### 6.1.2. Human Primary Hepatocyte-Derived Spheroids

A disc-shaped porous sponge was used as a scaffold for seeding human hepatocytes. These hepatic spheroids were shown to be capable of exhibiting liver functions, such as albumin synthesis, urea synthesis, and *CYP1A2* and *CYP3A4* transcript expression. These spheroids were susceptible to *P.*
*cynomolgi*, a causative agent of simian malaria, and *P. vivax*. Despite the complete development of *P. cynomolgi* from sporozoites to release erythrocyte-infectious merozoites from liver schizonts, the development of *P. vivax* sporozoites was not examined in this study [49]. A technical challenge in the use of 3D spheroids is the difficulty of assessing infectivity and parasitic load due to the limitations of confocal microscopy. Consequently, only confocal images at low magnification showed spheroids with fluorescence signals. Since cell size could not be examined, the relative fluorescence units were measured. Thus, it was difficult to observe small forms of the *P. vivax* parasite, which were presumably recognized as hypnozoites. Nevertheless, spheroid-infecting *P. vivax* parasites are sensitive to primaquine and autovaquone, and this porous sponge-based model allows the study of *P. cynomolgi*, a substitute model for the development of anti-hypnozoiticidal drugs for relapse prevention in humans.

### 6.2. Organoids

Organoids have emerged as a technological breakthrough and have been validated as a useful tool for studying tissue/organ development [50], disease modeling [51,52,53,54,55] and clinical applications [56,57]. According to a generally accepted definition, organoids are in vitro-grown cellular clusters or aggregates with the capability of proliferating, differentiating, 3D self-organizing and functioning similar to their tissue of origin and are thus often called miniature tissues. Organoids can be generated from adult stem cells isolated from primary tissues or pluripotent stem cells [58] (embryonic or iPSCs) (Figure 2). At present, organoids can be derived from the tongue, lung, mammary gland, liver, stomach, pancreas, intestine and prostate. Given the cell potency, various types of organoids are generated from pluripotent stem cells, including gastric, intestinal, hepatic, thyroid, lung, optic cup, cerebral, pituitary and inner organoids.

Intestinal organoids were first generated from intestinal stem cells [59], and endogenous intestinal stem cells, ECM and growth factors are crucial components of this process. Unlike in vitro 2D cultures, organoids are similar to primary tissue in both their composition and architecture, harboring small populations of self-renewing stem cells that give rise to fully differentiated progeny comprising all major cell lineages. Other advantages of organoids include their ability to be expanded indefinitely, cryopreserved as biobanks and easily manipulated using techniques similar to those established for traditional 2D monolayer culture. Thus, organoids represent an important bridge between traditional 2D cultures and in vivo mouse/human models [60], as they are more physiologically relevant than monolayer culture models and are far more amenable to the manipulation of niche components, signaling pathways and genome editing than in vivo models.

To explore whether organoids have potential for developing drugs targeting liver-stage malaria, we mainly discuss the key characteristics of liver/hepatic organoids. Although liver/hepatic organoids are reportedly generated from adult stem cells isolated from the liver, primary liver samples are hindered by their limited availability and uncertain quality. Importantly, pluripotent stem cells are capable of proliferating and differentiating into hepatic lineages, allowing the prescreening of specific phenotypes and the scaling up for high-throughput assays. Therefore, advances in generating liver/hepatic organoids from pluripotent stem cells are the main focus here.

To generate organoids from pluripotent stem cells, it is necessary to follow the development of the liver during embryogenesis. Briefly, after fertilization and the formation of germ layers, the endoderm gives rise to the hepatic endoderm, which then differentiates into hepatoblasts and functional hepatocytes. At present, stepwise protocols for the differentiation of hepatocytes from pluripotent stem cells involve establishment in a conventional 2D culture [30,61]. To form an organoid, a specific cell type is embedded in a gel-like ECM to allow cells to proliferate, differentiate and organize themselves in three dimensions. Two approaches are utilized to generate hepatic organoids from PSCs depending on the type of embedded cells.

#### 6.2.1. Single Cell Type-Derived Hepatic Organoids

In human fetal and neonatal livers, parenchymal cells expressing epithelial cell adhesion molecule (EpCAM) are capable of differentiating into hepatocytes and cholangiocytes (biliary epithelial cells) [62]. Similar to the fetal liver, a human adult liver has ductal cells expressing EpCAM. When cultured in 3D, the isolated EpCAM+ cells were shown to generate liver organoids [63] (Figure 2). To recapitulate in vivo findings, EpCAM-expressing endodermal cells were generated from human iPSCs and shown to self-organize into hepatic organoids in three dimensions [64]. By using this method, the hepatic organoids could be maintained in vitro for more than 16 months. An additional 10-14 days were required for the EpCAM-derived hepatic organoids to exhibit mature hepatocyte functions, including albumin synthesis, CYP3A4 activity, glycogen storage and low-density lipoprotein uptake [65]. While the long-term maintenance of EpCAM-derived hepatic organoids is known, the length of time that mature hepatic organoids can be cultured in vitro, a critical factor for testing antihypnozoite drugs, has not been determined. Our group recently reported the generation of hepatic organoids from human iPSC-derived endodermal cells expressing the *Plasmodium* sporozoite receptor CD81. Moreover, this hepatic organoid could be cultured for up to 60 days without the loss of albumin, CYP4A3 and CD81 expression at the protein level [65].

#### 6.2.2. Unknown Cell Type

Following the differentiation of human iPSCs into mature hepatocytes in 2D culture over 25 days, sphere-shaped structures floated over a monolayer of hepatocytes. When culturing the floating spheres in 3D culture using Matrigel, they were able to form spherical structures and exhibited the following characteristics of mature hepatocytes: albumin secretion, urea production and CYP3A4 activity. Since the expression levels of phase I drug-metabolizing enzymes and phase II detoxification enzymes in hepatic organoids were similar to those in the adult human liver, the hepatic organoids could be used to predict the toxicity of trogliazone (an antidiabetic drug) and acetaminophen. Notably, regarding the antimalarial drug primaquine, these hepatic organoids expressed CYP3A4 and CYP2C19 transcripts at levels similar to those in the adult liver. However, the expression of the CYP2D6 gene seemed to be low relative to that in the adult liver [56].

#### 6.2.3. Multiple Cell Types for Heterogenous Organoids

Given the heterogeneous population in the liver, the human iPSC-derived hepatic endoderm was cultured with mesenchymal stromal cells and endothelial cells on Matrigel covered with liquid medium to allow three-dimensional organization (Figure 2). The cells self-organized as aggregates, and microscopic observation revealed that the cellular architecture resembled a liver bud, forming liver tissue in the fetus. However, the iPSC-derived liver buds expressed a-fetoprotein during maintenance in vitro. Upon transplantation into mice, they were vascularized and could synthesize albumin, implying the requirement of some essential factors, which have not been explored [66,67]. Our group applied the human iPSC-derived liver bud for the treatment of vivax malaria. By day 6 post inoculum, transcripts of Plasmodium 18S rRNA and merozoite surface protein 1 were detected (Figure 3, unpublished data), implying the susceptibility of this 3D liver model.

## 7. Opportunity for the Use of Organoids in Drug Development for Liver-Stage Human Malaria

Organoids were employed to model host–microbe interactions for *Helicobacter pylori* [68,69] and Shiga toxin-producing *Escherichia coli* [70]. Some factors must be considered before hepatic organoids can be utilized, including determining whether a long-term culture causes loss of hepatocyte function and determining their ability to metabolize drugs. Here, evidence that hepatic organoids are superior to the 2D conventional culture is highlighted. First, Takayama et al. showed that responses to drug toxicity in 3D models were similar to those of their in vivo counterparts [71]. Moreover, many side effects of drugs lead to acute liver injury. Thus, hepatic organoids could be utilized to predict the in vivo liver toxicity of drugs or small molecules before commencing expensive clinical trials [72]. Recently, the gene expression profile of human iPSC-derived hepatic organoids was shown to more closely resemble that of the adult liver than that of 2D cultured hepatocytes derived from the same source [56].

Here, a method for using hepatic organoids or liver bud-like clumps is proposed (Figure 4). As illustrated in Figure 4A, the hepatic organoid comprises a hollow sphere (yellow-colored balls) embedded in a semisolid gel (pink) and covered with liquid medium (orange). Following sporozoite inoculation in liquid medium, sporozoites must penetrate into the semisolid gel. The point at which the inoculated sporozoites are able to reach the organoid needs to be optimized depending on the thickness of the gel and the organoid. Thus, the semisolid gel acts as a barrier to parasite invasion. Moreover, the complete development of liver-stage malaria is determined by the generation of liver schizonts capable of releasing erythrocyte-infectious merozoites. Thus, adding human erythrocytes to a 2D culture of hepatocytes is simple due to direct contact between the released merozoites and erythrocytes. By contrast, where merozoites are released in the 3D hepatic organoid, whether they float in the hollow space or are trapped in the semisolid gel, must be examined. Given that the gel can change to a liquid form, the 3D hepatic organoid structure can be dissociated to release merozoites. For the liver bud-like model (Figure 4B), cell clumps were directly exposed to liquid medium, allowing parasite invasion and merozoite release. Thus, assessment of *Plasmodium* development in liver bud-like cell clumps is possibly simpler than that of the former model, in which the semisolid gel acts as a barrier.

## 8. Challenges in the Preclinical Use of Organoids as a Liver-Stage Malaria Model

Although the development of organoid models represents a major technological breakthrough, the limited presence of stromal components, including immune cells, limits their use in modeling inflammatory responses to infection or drugs. Another obstacle for the use of organoids in drug screening is the limited drug penetration resulting from the relatively rigid ECM [73,74], which could be addressed by varying the composition of the ECM. Moreover, organoid cultures are often intrinsically heterogeneous in terms of their viability, size and shape [75], which complicates the analysis of drug toxicity and efficacy. To assess drug efficacy, *Plasmodium* development into liver-stage schizonts is used as a proxy for parasite growth. Thus, liver-stage schizonts in hepatocytes are primarily quantified using antibody-based assays, i.e., a polyclonal antibody specific for *Plasmodium* HSP70 or a monoclonal anti-*P. falciparum* circumsporozoite protein antibody [14]. Following the antigen-antibody reaction and fluorescence signal detection, the microscopic images are used to manually count the antibody-bound cells. Hence, this method is prone to have low sensitivity and to be subjective and laborious, hindering high-throughput screening. Given the advances in cell visualization technology, counting based on high-content images is faster and less subjective than manual cell counting [17]. Nevertheless, the multistep cell fixation and antibody binding procedure is a hindrance to the potential use of antibody-based assays. Several reports have utilized transgenic strains of Plasmodium parasites capable of emitting fluorescence signals. For example, a bioluminescent *P. yoelii*, a causative agent of rodent malaria, was reportedly used as a readout of parasite infection and growth [22]. Notably, human and nonhuman malaria parasites exhibit different genome organizations [76] and transcriptomes [77], requiring the translation of data for the human context.

## 9. Conclusions

Despite some asymptomatic cases, liver-stage malaria is a prophylactic and antirelapse target. All of the available drugs targeting liver-dwelling *Plasmodium* cause lethal side effects in subjects with glucose-6-phosphate dehydrogenase deficiency. Progress on new drug development is slow relative to that for drugs targeting intraerythrocytic parasites. In this review, we proposed a strategy for overcoming the following long-standing hurdles related the development of new drugs based on in vitro models: nonphysiological relevance, batch-to-batch variation, shortages and ethical issues. Given the potential of pluripotent and adult stem cells in modelling diseases in a physiological context, the organoid is emerging as a next-generation model of liver-stage malaria. However, sporozoite invasion, the assessment of parasite growth, variations in size and shape, and drug penetration are major challenges in the use of organoids to investigate liver-stage malaria. Nevertheless, advances in stem cell technologies and multidimensional culture may overcome these limitations, leading to a next-generation model as a supportive or alternative method for the preclinical phase of drug discovery.

## Figures and Tables

**Figure 1 antibiotics-10-00642-f001:**
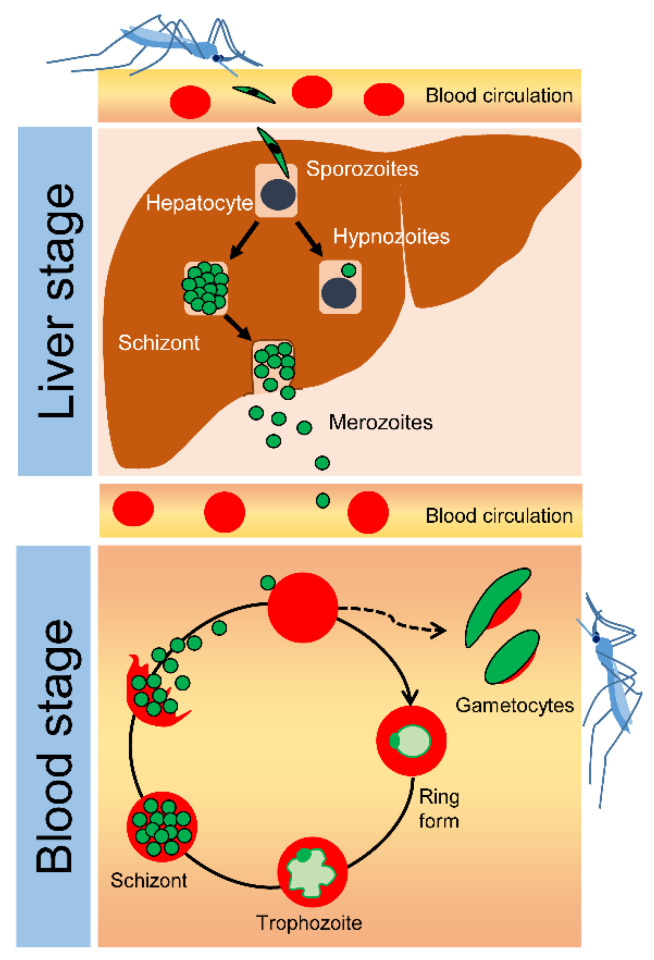
Development of *Plasmodium* parasites in the human body [7]. During blood feeding, a female mosquito secretes anti-blood clotting factors from the salivary gland, from which sporozoites are collaterally released into the bloodstream. Upon inoculation through the dermis, sporozoites eventually invade the liver, the first location of *Plasmodium* infection in humans. In the invaded liver, sporozoites enter hepatocytes and undergo asexual multiplication (schizogony), resulting in a merozoite-harboring hepatocyte known as a liver schizont. After approximately one week, the hepatic schizonts burst, and thousands of merozoites enter the bloodstream. For *P. vivax* and *P. ovale*, some merozoites undergo a stage of inactive proliferation (dormancy), known as hypnozoites, which cause relapse months or years later. Hepatocyte-derived merozoites invade erythrocytes and begin a repeated cycle of asexual, intraerythrocytic development involving ring-formed trophozoites, trophozoites, and schizonts. Rupture of schizonts causes periodic fever and, for some *Plasmodium* species, possibly leads to severe, lethal symptoms, such as anemia, multiorgan failure and cerebral malaria. Some host erythrocytes develop into sex cells, termed gametocytes. Female *Anopheles* mosquitoes take up gametocytes during a blood meal. In the mosquito gut, fertilization of gametocytes results in the production of sporozoites, which specifically migrate to the salivary glands of *Anopheles* mosquitoes to await inoculation at the next blood feed.

**Figure 2 antibiotics-10-00642-f002:**
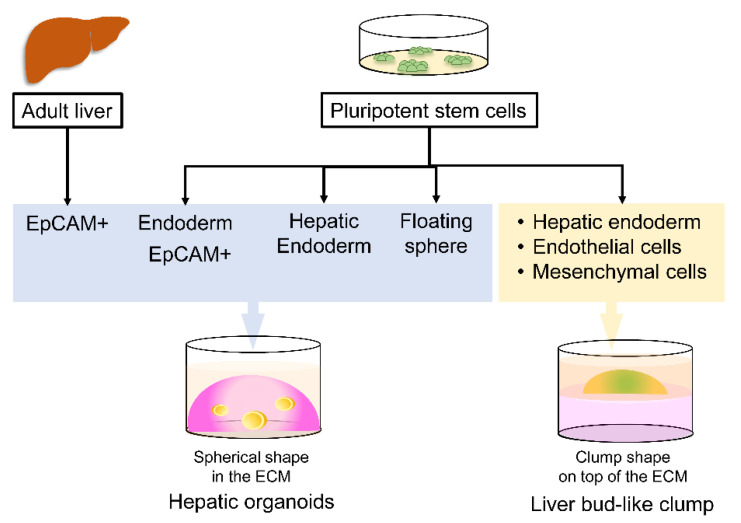
Source of liver/hepatic organoid-initiating cells. In the adult liver, EpCAM-expressing cells are capable of forming hollow, spherical shaped cells when embedded in a gel-like substance composed of the ECM and covered with liquid medium supplemented with essential growth factors and nutrients. Adult primary cell-derived hepatic organoids exert liver functions, including albumin synthesis, glycogen and lipid storage, and cytochrome P450 activity. As an alternative to the adult liver, human PSCs have been employed for the preparation of EpCAM+ endodermal, hepatic endodermal, endothelial and mesenchymal cells. As their counterparts in the adult liver, EpCAM+ endodermal cells and the hepatic endoderm formed hepatic organoids. In a 2D culture of hepatocytes derived from human PSCs, floating spheres appear over monolayers and are capable of forming 3D hepatic organoids. The liver bud, a clumped shape lacking a hollow space, forms upon the plating a mixture of hepatic endodermal, endothelial and mesenchymal cells on Matrigel. To date, the PSC-derived liver bud is the most physiologically relevant structure to the adult liver in terms of its architecture and function.

**Figure 3 antibiotics-10-00642-f003:**
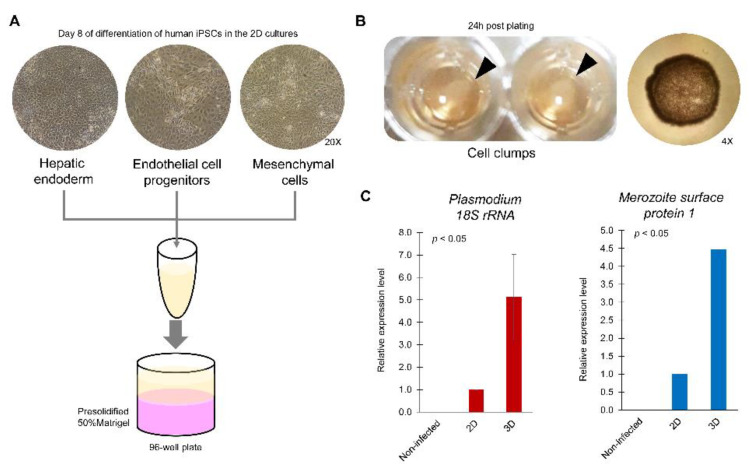
Potential of liver bud-like cell clumps to model liver-stage malaria. (**A**) Protocol to generate hepatic endodermal, endothelial and mesenchymal cells from the human iPSC MUi019 line. Cells were harvested from a monolayer of 2D culture and mixed in culture medium. The cell mixture was then added to a presolidified 50% Matrigel (pink) and allowed to settle on the surface. (**B**) At 24 h of cell plating, the cells had shrunk and became clumped (arrowheads). Microscopic observation revealed the round shape of a cell clump (4× objective lens). (**C**) At day 6 post inoculation of *P. vivax* sporozoites, the mRNA expression of Plasmodium 18S rRNA and merozoite surface protein 1 in the 3D culture of liver bud-like cell clumps was higher than that in the 2D culture of human hepatocytes derived from the human iPSC MUi019 line.

**Figure 4 antibiotics-10-00642-f004:**
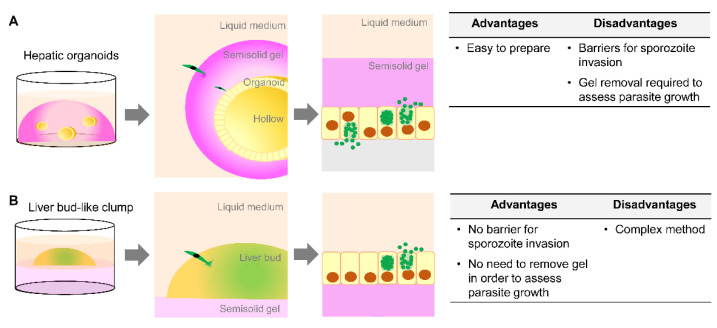
Proposed methods for the use of hepatic organoids or liver bud-like clumps. (**A**) Hepatic organoids (yellow-colored balls) are embedded in a semisolid gel (pink dome-shape) and covered with liquid medium (orange). Following sporozoite inoculation in liquid medium, sporozoites penetrate into the semisolid gel and reach the organoid. Subsequently, mature liver schizonts release erythrocyte-infectious merozoites in two ways: (1) floating in the hollow space (gray area) or trapping in the semisolid gel. (**B**) Liver bud-like cell clumps are directly exposed to liquid medium, allowing parasite invasion and the release of merozoite. The advantages and disadvantages of both of the proposed methods are summarized.

**Table 1 antibiotics-10-00642-t001:** Liver-like models applied in preclinical studies of human malaria.

Models		Infection Rate (%) Based on		Merozoites		Hypnozoites ^†^ (dpi)	Applications	References
		Sporozoites *	Hepatocytes	Detection (dpi)	Infectivity			
Hepatocellular carcinoma cell lines	HepG2-A16	0.0001 (P.v.)	N/A	9	N/A	Yes (5–15)	Development	[13]
		0.4–2.5 (P.f.)	N/A	N/A	N/A	N/A	Sporozoite invasion	[15]
	HHS-102	N/A	0.009 (P.f.)	12–13	Yes	Not applicable	Development	[14]
Primary human hepatocytes	Mazier et al., 1985	N/A	N/A	12–13	Yes	Not applicable	Development	[16]
	March et al., 2013	0.03 (P.f.) 0.013 (P.v.)	0.18 (P.f.)	6–10	Yes (P.f.)	Yes (up to 21)	Development Vaccination Drug testing and screening	[17]
	Roth et al., 2018	0.6–2 (P.f.) 2–8.3 (P.v.)	N/A	7–8 (P.f.) 9–11 (P.v.)	Yes	Yes (6–8)	Development Sporozoite invasion Drug testing and screening	[18]
Immortalized cells	HC-04	N/A	0.066 (P.f.) 0.041 (P.v.)	7 (P.f.) 10 (P.v.)	Yes Yes	Yes (28)	Development Sporozoite invasion	[19] [20]
	imHC	0.14 ± 0.16 (P.v.)	N/A	10 (P.v.)	N/A	Yes (14)	Development Drug testing	[21]
Pluripotent stem cells	Hepatocytes	N/A	N/A	6 (P.f.) 8 (P.v.)	N/A	N/A	Development Drug testing	[22]

* Number of sporozoite-infected hepatocytes/total number of viable, inoculated sporozoites. ^†^ Nondividing, small-form parasites presumably recognized as hypnozoites. Dpi, day post inoculation; N/A, not determined; P.f., *P. falciparum*; P.v., *P. vivax.*

**Table 2 antibiotics-10-00642-t002:** In vitro models for plasmodicidal activity assays against human liver-stage malaria.

Cell type	ECM	Well Format	Culture	Drugs	Biomarkers	References
Primary human hepatocytes (PHHs)	Type I collagen (rat tail)	96-well plates	Coculture with murine embryonic fibroblasts	Primaquine	Circumsporozoite protein (P.f.)	[17]
		384-well plate	Monoculture	Compound library Phosphatidylinositol 4-kinase inhibitor (KDU691) Primaquine Tafenoquine Atovaquone	GAPDH (P.f.) UIS4 (P.v.)	[18]
imHCs	Matrigel	8-well plate	Monoculture	Primaquine	*Plasmodium* HSP70 UIS4 (P.v.)	[21]
iPSC-derived hepatocytes	No information	No information	Monoculture	Primaquine	HSP70 (P.f.)	[22]

P.f., *P. falciparum*; P.v., *P. vivax*.

## Data Availability

Not applicable.

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
