# Peer review of "Next-Generation Human Liver Models for Antimalarial Drug Assays"

_antibiotics, 2021, doi:10.3390/antibiotics10060642_

Round 1
Reviewer 1 Report
The article is very interesting. However, before it could be considered for publication, authors need to incorporate suggestions of the reviewer and extensively revise their manuscript.
General comments formatting of the contents need to be done as per guidelines of the journal.
Sentence formation needs crosscheck. Grammatical mistakes need to be minimized.
Abstract section does not give proper information. Abstract means a full-fledged summary that should give readers highlights of the information and topics covered in the manuscript. Please revise the abstract.
Section Introduction
Authors need to give a background of malaria. The authors are advised to cut short the introduction and confine it to what is concerned to the contents of the manuscript.
Section 2
Long paras need to be avoided to avoid any confusion between the statements. References need to be crosschecked as only 3 references are given for the contents of this section.
Section 3
I fail to understand as why reference was added to sub-heading. Authors can summarize the information pertaining to this reference in the following sentence. Again, only 3 references for the information.
Section 4
In vitro word needs to be italic even in the abstract section.
HepG2-A16 cells 13 …It is better to put reference 13 in correct format.
Long sentences need to be avoided.
Table 1
It is better to put application column at the end of the table if it applies to both columns on right as well as left to it.
Reference’s column can be added separately as it mismatches at different locations in the table. E.g., Primary human hepatocytes under model heading. Avoid confusion.
Table 2
It is better to put references only in one format (either by author names or by numbering). The same case is also present at other places of the manuscript.
Section 6
Again, references added to sub-headings 6.1.1 and 6.1.2 and no references to information given under these headings. Unwarranted statements should need to be minimized.
Again, reference to 6.2.2 with unknown cell type subheading
Section 8
Only 3 references. References for information to different contexts needs to be added.
Section conclusion
The section needs to be a bit more elaborative and should highlights importance of the study and future directions with possible limitations.
Author Response
May 26th, 2021
Dear Editors
We thank the reviewers for their generous comments on the revised manuscript and have edited the revised manuscript to address their concerns.
In particular, we have responded point-by-point to reviewer’s comments below, with our responses immediately below the comment in blue color. The revised portions are indicated in blue color and track changes in the revised manuscript. Moreover, English language in the revised manuscripts was re-edited by American Journal Experts (the certificate attached).
We believe that the revised manuscript is now suitable for publication in Antibiotics.
Yours sincerely,
Kasem Kulkeaw, Ph.D.
Responses to reviewers
Reviewer 1:
The article is very interesting. However, before it could be considered for publication, authors need to incorporate suggestions of the reviewer and extensively revise their manuscript.
General comments formatting of the contents need to be done as per guidelines of the journal.
Sentence formation needs crosscheck. Grammatical mistakes need to be minimized.
Abstract section does not give proper information. Abstract means a full-fledged summary that should give readers highlights of the information and topics covered in the manuscript. Please revise the abstract.
Response to reviewer:
Firstly, I am highly appreciate your time and grateful to thanks for your valuable supportive comments, which enable improvement of the review. Please find my responses following each comments.
As you suggested, the revised MS was sent to the American Journal Experts to edit English (Please find the attached certificate).
The improvement of the abstract now covers key contents and topics of the MS (line 21-25).
1.1 Section Introduction
Authors need to give a background of malaria. The authors are advised to cut short the introduction and confine it to what is concerned to the contents of the manuscript.
Response to reviewer:
To shorten and confine introduction, some sentences were removed. In the revised MS, the changes are in line 59-62.
The background of malaria is provided in line 66-70, 76-78, 83, 85-87 and figure legend 1 (482-497).
1.2 Section 2
Long paras need to be avoided to avoid any confusion between the statements. References need to be crosschecked as only 3 references are given for the contents of this section.
Response to reviewer:
I sincerely apologies for confusion caused by long paras. Changes are in line 66-68, 76-78, and 85-87.
Since this section highlights the unfilled gaps in hepatic malaria, and there two review articles comprehensively provide details of liver invasion mechanism, thus the revised MS will not cover the same content.
1.3 Section 3
I fail to understand as why reference was added to sub-heading. Authors can summarize the information pertaining to this reference in the following sentence. Again, only 3 references for the information.
Response to reviewer:
Thanks for pointing out. It was typos for references. Now the reference no. 6 is in line 109.
There are four references in this section. Reference no. 6 was added in line 109 and 124.
1.4 Section 4
In vitro word needs to be italic even in the abstract section.
HepG2-A16 cells 13 …It is better to put reference 13 in correct format.
Long sentences need to be avoided.
Response to reviewer:
I sincerely apologies for several typos.
Regarding reference no. 13, the format and some corrections were made (line 134-138).
All in vitro words are now in italic.
I carefully read every sentences, especially the long one, and hope that the inserted commas could aid understand of readers.
1.5 Table 1
It is better to put application column at the end of the table if it applies to both columns on right as well as left to it.
Reference’s column can be added separately as it mismatches at different locations in the table. E.g., Primary human hepatocytes under model heading. Avoid confusion.
Response to reviewer:
As you suggested, the table 1 was edited (line 146-151).
1.6 Table 2
It is better to put references only in one format (either by author names or by numbering). The same case is also present at other places of the manuscript.
Response to reviewer:
References are written as numbers (line 233-249).
1.7 Section 6
Again, references added to sub-headings 6.1.1 and 6.1.2 and no references to information given under these headings. Unwarranted statements should need to be minimized.
Again, reference to 6.2.2 with unknown cell type subheading
Response to reviewer:
I carefully read the statements in section 6, and hope that all of them are reasonable. However, this would be helpful, if the unwarranted statements are pointed out. So that, I shall minimize them.
References were removed from sub-heading of 6.1.1, 6.1.2 and 6.2.2 to the line 307, 315 and 393, respectively.
1.8 Section 8
Only 3 references. References for information to different contexts needs to be added.
Response to reviewer:
More references were added (line 443, 445, and 454).
1.9 Section conclusion
The section needs to be a bit more elaborative and should highlights importance of the study and future directions with possible limitations.
Response to reviewer:
The conclusion was elaborated by highlighting potential of organoid and emphasizing its limitations (Line 470-475).
Reviewer 2 Report
The review is outstanding.
I found minor typos, like in line 140, 163 (the manner in which the references are added), line 287 (P.cynomolgy).
In page 2, line 33-34 the sentence should be reformulated as it advances the idea that only P.falciparum and P.vivax are causing malaria.
Author Response
May 26th, 2021
Dear Editors
We thank the reviewers for their generous comments on the revised manuscript and have edited the revised manuscript to address their concerns.
In particular, we have responded point-by-point to reviewer’s comments below, with our responses immediately below the comment in blue color. The revised portions are indicated in blue color and track changes in the revised manuscript. Moreover, English language in the revised manuscripts was re-edited by American Journal Experts (the certificate attached).
We believe that the revised manuscript is now suitable for publication in Antibiotics.
Yours sincerely,
Kasem Kulkeaw, Ph.D.
Reviewer 2
The review is outstanding.
I found minor typos, like in line 140, 163 (the manner in which the references are added), line 287 (P.cynomolgy). In page 2, line 33-34 the sentence should be reformulated as it advances the idea that only P.falciparum and P.vivax are causing malaria.
Response to reviewer:
Firstly, I would like to thank for your time and valuable comments. I sincerely apologies for several typos and did corrections accordingly.
- Regarding reference no. 13, the format and some corrections were made (line 134-138).
- To my understanding, reference no. 23, 24, and 25 are added automatically by Endnote.
- cynomlgi changed to P. cynomolgi (line 312).
- For page 2 (line 33-34), the sentence was reformulated as shown below (page 2, line 35-36 in the revised MS), “The causes of malaria are the protozoan parasites, including Plasmodium falciparum, vivax, P. ovale, P. malariae and P. knowlesi. P. falciparum is the most virulent and dominant species, and P. vivax, which has the widest geographical distributions and exert a major economic impacts.”

Reviewer 3 Report
Prof. Dr. Nicholas Dixon
Editor-in-Chief
Antibiotics
Dear Editor
I have the following comments on the article: “Next-generation human liver models for antimalarial drug assays”.
Reviewer Comments,
This paper described the potential of pluripotent and adult stem cells to assess the efficacy of hepatic plasmodicidal activity. The following should be considered by authors to improve the quality of the manuscript:
Comments:
- Please improve the language aspects of the manuscript as there are grammatical errors and typos.
The manuscript has some problems with English usage and grammar.
-In discussion could include the findings made by Meister et al., 2011 (Imaging of Plasmodium Liver Stages to Drive Next-Generation Antimalarial Drug Discovery); Brunschwig et al., 2018 (UCT943, a Next-Generation Plasmodium falciparum PI4K Inhibitor Preclinical Candidate for the Treatment of Malaria); and Antonova-Koch et al., 2018 (Open-source discovery of chemical leads for next-generation chemoprotective antimalarials).
By signing this letter, I approve that this article be accepted in the present form.
Author Response
May 26th, 2021
Dear Editors
We thank the reviewers for their generous comments on the revised manuscript and have edited the revised manuscript to address their concerns.
In particular, we have responded point-by-point to reviewer’s comments below, with our responses immediately below the comment in blue color. The revised portions are indicated in blue color and track changes in the revised manuscript. Moreover, English language in the revised manuscripts was re-edited by American Journal Experts (the certificate attached).
We believe that the revised manuscript is now suitable for publication in Antibiotics.
Yours sincerely,
Kasem Kulkeaw, Ph.D.
Reviewer 3:
Reviewer Comments,
This paper described the potential of pluripotent and adult stem cells to assess the efficacy of hepatic plasmodicidal activity. The following should be considered by authors to improve the quality of the manuscript:
Comments:
- Please improve the language aspects of the manuscript as there are grammatical errors and typos.
The manuscript has some problems with English usage and grammar.
Response to reviewer:
Firstly, I am highly appreciate your time and grateful to thanks for constructive feedbacks. To improve the English usage and grammar, the revised MS was send to the American Journal Experts to edit language. Please find the certificate of the American Journal Experts.
-In discussion could include the findings made by Meister et al., 2011 (Imaging of Plasmodium Liver Stages to Drive Next-Generation Antimalarial Drug Discovery); Brunschwig et al., 2018 (UCT943, a Next-Generation Plasmodium falciparum PI4K Inhibitor Preclinical Candidate for the Treatment of Malaria); and Antonova-Koch et al., 2018 (Open-source discovery of chemical leads for next-generation chemoprotective antimalarials).
Response to reviewer:
Given that this review focus on liver-stage model, a paragraph of the suggested references was added in the heading of “Current toolbox for assaying the plasmodicidal activity of liver-stage malaria” of the revised MS (line 264-277).